# Exploring the Concept of Lineage Diversity across North American Forests

**Kyle G. Dexter** [1,2,]*[ ], **Ricardo A. Segovia** [1,3][ ] **and Andy R. Griffiths** [1]

1   School of GeoSciences, University of Edinburgh, Edinburgh EH9 3FF, UK; segoviacortes@gmail.com (R.A.S.);
    andy.griffiths@ed.ac.uk (A.R.G.)
2   Royal Botanic Garden Edinburgh, Edinburgh EH3 5LR, UK
3   Instituto de Ecología y Biodiversidad, Santiago 7800003, Chile
*   Correspondence: kyle.dexter@ed.ac.uk; Tel.: +44-(0)-131-650-7439

**Abstract:** Lineage diversity can refer to the number of genetic lineages within species or to the number of deeper evolutionary lineages, such as genera or families, within a community or assemblage of species. Here, we study the latter, which we refer to as assemblage lineage diversity (ALD), focusing in particular on its richness dimension. ALD is of interest to ecologists, evolutionary biologists, biogeographers, and those setting conservation priorities, but despite its relevance, it is not clear how to best quantify it. With North American tree assemblages as an example, we explore and compare different metrics that can quantify ALD. We show that both taxonomic measures (e.g., family richness) and Faith's phylogenetic diversity (PD) are strongly correlated with the number of lineages in recent evolutionary time, but have weaker correlations with the number of lineages deeper in the evolutionary history of an assemblage. We develop a new metric, time integrated lineage diversity (TILD), which serves as a useful complement to PD, by giving equal weight to old and recent lineage diversity. In mapping different ALD metrics across the contiguous United States, both PD and TILD reveal high ALD across large areas of the eastern United States, but TILD gives greater value to the southeast Coastal Plain, southern Rocky Mountains and Pacific Northwest, while PD gives relatively greater value to the southern Appalachians and Midwest. Our results demonstrate the value of using multiple metrics to quantify ALD, in order to highlight areas of both recent and older evolutionary diversity.

**Keywords:** temperate forests; species richness; assemblage lineage diversity; phylogenetic diversity; evolutionary diversity; United States; trees; TILD

## 1. Introduction

The evolutionary lineage is a fundamental concept in biology, denoting a group of organisms connected by ancestor-descendent relationships [1]. Evolutionary lineages are hierarchically structured; multiple younger evolutionary lineages can be nested within an overarching older lineage, or clade. Thus, multiple genetically diverged lineages can exist within a single taxonomic species, and multiple species can belong to older evolutionary lineages, such as genera, families or orders. Knowing the number of lineages in different ecological assemblages and biogeographic regions gives insights into evolutionary process, biogeographic history, and conservation priorities. For example, an assemblage or region that houses many lineages can be interpreted as having a richer evolutionary history, and therefore may be a greater priority for conservation than one that houses few. However, the conservation value of lineage diversity has yet to be fully, and persuasively, communicated [2–4]. Providing clear and accurate quantification of lineage diversity may assist its integration into conservation practice.

In its most basic form, quantifying the number of lineages in assemblages could consist of counting the number of species. However, the term lineage diversity is generally applied when the units are not species, but a shallower or deeper evolutionary level, i.e., within or above the species taxonomic rank (see [5–9] for examples below species rank; see [10–14] for examples above species rank). In this paper, we focus on lineage diversity above the species rank. Employing tree assemblages in the contiguous United States, we explore various metrics by which assemblage lineage diversity (hereafter ALD) might be quantified, using taxonomic and phylogenetic approaches. Given its pertinence to conservation prioritisation, we focus specifically on the richness dimension of ALD.

Taxonomy is a hierarchical system for organising biological diversity. As such, it provides an apparently straightforward means of quantifying ALD at different evolutionary depths, for example by tallying the number of genera, families or orders in assemblages. However, Linnean taxonomic ranks are not 'natural' in the sense that they do not directly correlate to any precise evolutionary age. Some clades of a given taxonomic rank may actually be younger than clades of a putatively lower taxonomic rank. For example, the genus *Pinus* (Pinaceae) may be as old as 100 million years [15], which is older than most angiosperm families [16]. If one were to compare an assemblage of four *Pinus* species with an assemblage of four angiosperm species belonging to different genera in the same family, and ALD were estimated as the number of genera in each assemblage, the angiosperm assemblage would appear to have 4x higher ALD. However, all four species in the assemblage of *Pinus* may have diverged from each other prior to the age of the most recent common ancestor of the four species in the angiosperm assemblage (similar to mock assemblages B and C in Figure 1), which could mean that the assemblage of *Pinus* has greater conservation value because it encompasses greater total evolutionary history, in terms of time or branch lengths.

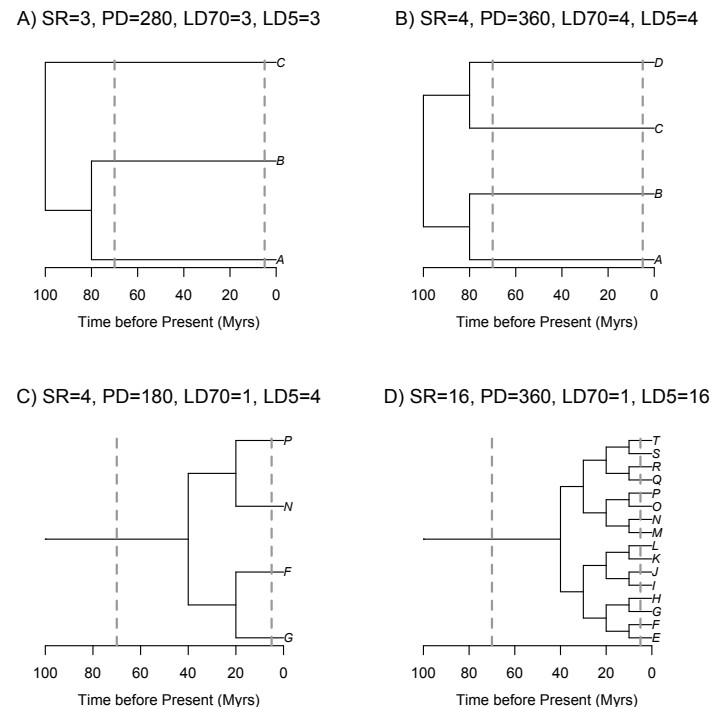

**Figure 1.** Example phylogenies for four mock assemblages (**A–D**) with contrasting species richness (SR), phylogenetic diversity (PD) and phylogenetic structure (LD70 = number of lineages 70 Ma; LD5 = number of lineages 5 Ma).

The advent of molecular phylogenetics has allowed researchers to move past taxonomic approaches to quantifying ALD. Using a temporally calibrated phylogeny, one can choose a certain evolutionary age–say X millions of years (Myrs)–and then readily estimate the number of lineages at

X million years ago (Ma) in an assemblage of species. Further, one could examine how the number of lineages varies at different time slices across a set of assemblages, or geographic space (*sensu* Jønsson et al., 2011). This is directly analogous to constructing a lineage through time plot for a given evolutionary clade [17], and indeed, studies have proposed constructing lineage through time plots for individual communities or assemblages [18]. However, it is not clear at which evolutionary age, or phylogenetic depth, one should be counting lineages. An assemblage that has more lineages than another assemblage at one, deeper time slice might have fewer lineages at a more recent time slice (compare assemblages B vs. D in Figure 1), which could be driven by variation in diversification histories, community assembly, or numerous other processes. It would be ideal to have metrics for ALD that integrate over the evolutionary history of the clade being studied.

Faith (1992) [19] developed a simple metric, phylogenetic diversity (PD), to estimate the evolutionary history present in communities or assemblages of species, which is calculated by summing the length of all branches in a phylogeny that includes all taxa present in an assemblage, and only those taxa. While this metric is related to the number and age of evolutionary lineages present in an assemblage, and thus may serve as a proxy for ALD, Figure 1 demonstrates that inferences of ALD based on calculating PD may not always be straightforward. In this contrived scenario, it seems clear that Assemblage A has less ALD than Assemblage B and that Assemblage C has less ALD than Assemblage D. The calculations of PD, and even species richness, would support this visual observation. Further, it seems plausible that Assemblage A has more lineage diversity than Assemblage C, even though Assemblage C has more species. However, do Assemblages B and D really have identical ALD even though they have such a discrepancy in species richness? Comparing Assemblages B and D is challenging because they have such different phylogenetic structures. Assemblage B has 4x as many lineages at 70 Ma, while Assemblage D has 4x as many lineages at 5 Ma. For this reason, researchers have suggested that the amount of PD an assemblage contains above or below that expected given its SR is a better measure of ALD [12,13]. However, if we were to follow that approach, then Assemblage C might be considered to have more ALD than Assemblage D (its ratio of PD:SR is twice that of Assemblage D), even though at all phylogenetic depths Assemblage D has the same or more lineages than Assemblage C. Clearly, more work is needed to determine which metrics derived from phylogenies may provide the best measures of ALD that integrate over evolutionary timescales.

The overarching goal of the present manuscript is to explore the behaviour of different metrics that may potentially be used to quantify ALD. As our empirical example, we focus on tree assemblages in the contiguous United States. These tree assemblages provide an ideal system for such an empirical study, as over 150,000 forest inventory plots have been sampled in a standardised way by the U.S. National Forest Service, and existing time-calibrated phylogenies encompass nearly all species present in the plots. We use this large dataset to (1) quantify the ALD using different taxonomic and phylogenetic metrics; (2) assess the relationship of different metrics with each other and with the number of lineages at different evolutionary depths; and (3) map variation in ALD across the contiguous United States. To give context to our results, we conduct a clustering analysis of assemblages based on their shared evolutionary history, thereby determining the main evolutionary groups of tree assemblages in the contiguous United States.

## 2. Materials and Methods

### 2.1. Data Sources

We accessed compositional data from 177,549 plots sampled across the contiguous United States by the Forest Inventory and Analysis (FIA) Program of the U.S. Forest Service [20], via the BIEN package [21] for the R Statistical Environment [22]. The FIA protocol records trees $\geq$12.7 cm diameter at breast (dbh) in four 168.3 m$^2$ subplots that are 36.6 m apart. The main evident spatial data gaps in this dataset are the state of Louisiana and the eastern part of the state of Kentucky.

In order to obtain a phylogeny that covered all species in the FIA tree plot inventory dataset, we combined the temporally calibrated ultrametric phylogenies for North American gymnosperm and angiosperm trees from Ma et al. (2016) [23] (see Figure 2). We set the age of the split between angiosperms and gymnosperms at 350 Ma [24]. After resolving synonyms according to The Plant List (2013), version 1.1 (http://www.theplantlist.org/, accessed in December 2018), we manually added the tree species present in the FIA dataset, but absent in the phylogeny. Their exact placement was based on consultations of the systematics literature (see Table S1 for species added and associated literature reference), with the added taxon being placed halfway along the branch leading to its sister species or clade in the phylogeny. The branch length leading to the added taxon was set to a value such that the tree remained ultrametric. The phylogeny file used in this study is available in Appendix B.

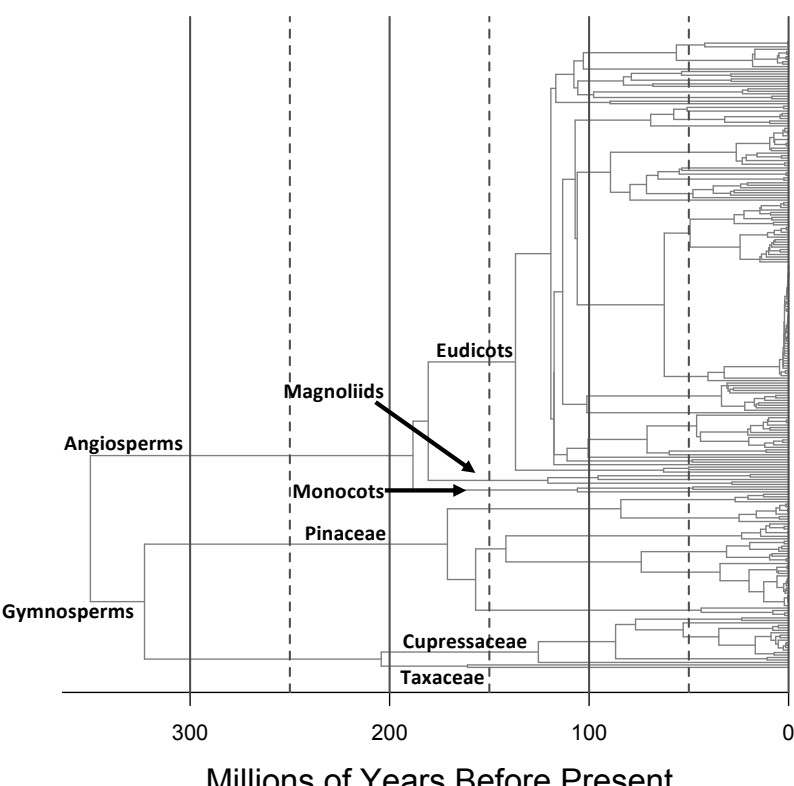

**Figure 2.** Phylogeny of all tree species present in the contiguous United States in the US Forest Inventory and Analysis (FIA) dataset, based on the phylogenies for gymnosperms and angiosperms in Ma et al. (2016).

## 2.2. Assemblage Lineage Diversity (ALD) Metrics

### 2.2.1. Taxonomic Measures

In the absence of phylogenetic data, the number of supraspecific lineages in assemblages can be calculated as the number of taxa of a higher taxonomic rank. Classification systems are consistent across angiosperms and gymnosperms up to the order level, and we therefore tabulated the following taxonomic measures of lineage diversity for assemblages: **number of genera, number of families and number of orders**. The taxonomy table is available in Appendix C.

### 2.2.2. Phylogenetic Measures

Since the advent of molecular phylogenetics, diverse metrics have been developed and implemented to quantify the lineage, or evolutionary, diversity of assemblages from phylogenies

(e.g., References [25–27]). We focus here on metrics that aim to quantify the 'richness dimension of phylogenetic diversity' [28], as our interest is in 'how much' lineage diversity is in assemblages, not how diverged lineages are from each other (e.g., as quantified by mean pairwise phylogenetic distance) or how evenly lineages are represented (e.g., as quantified by phylogenetic species evenness; [29]). In addition, conservation prioritisation is generally based on which species are present, not their relative abundance (which could reflect disturbance histories or other idiosyncratic processes), and we therefore focus on presence/absence metrics. This also increases the general utility of our approach, as abundance information is not available for many datasets.

We started by calculating the most basic metric of ALD, **phylogenetic diversity**, or **PD** [19], which is the sum of all branch lengths in each assemblage, including the branch that goes to the root of all seed plants. We also include its estimate standardised for variation in species richness. This is accomplished by calculating the first two moments of the null expectation for PD, given the number of species in the assemblage, and using them to calculate a standardised effect size. The moments of the null distribution can be calculated by randomly shuffling the tips of the phylogeny many times, but there is an analytical expectation for these moments, which is the approach we used [30]. We refer to this metric as the **standardised phylogenetic diversity**, or **sPD**.

We also calculated two additional proposed measures of the richness dimension of phylogenetic diversity, the **phylogenetic species richness**, or **PSR** [29] and the sum of **evolutionary distinctiveness**, or **sumED** [31]. PSR can essentially be considered a measure of species richness that takes into account the phylogenetic relatedness of taxa in an assemblage. If the assemblage is composed entirely of closely related species, this will produce a lower value of PSR than if the assemblage were composed of distantly related taxa. In practice, this measure is obtained by multiplying the mean pairwise phylogenetic distance between species in an assemblage by its species richness (and dividing by two, so that it represents distance to tips from the most recent common ancestor for each pair of species). For sumED, we first calculated the evolutionary distinctiveness of each species in our dataset, based on the entire phylogeny representing all species, following the fair proportions approach of Isaac et al., (2007) [32]. This is essentially a measure of how phylogenetically isolated each species is, relative to the given phylogeny. We then summed the evolutionary distinctiveness values for the species in each assemblage, following Safi et al., (2013) [31].

As our overarching goal in this study was to quantify ALD over the full evolutionary time of the clade of study (here, seed plants), we developed an additional metric that may better capture this, which we term **time integrated lineage diversity**, or **TILD**. If one constructs a lineage through time (LTT) plot for each assemblage (*sensu* Yguel et al., 2016) [18], one can simply integrate the area under this curve as a measure of the total lineage diversity of the assemblage over time. This integral is mathematically identical to the phylogenetic diversity of the assemblage, when including the root branch. However, in considering an LTT plot built from extant species, as LTT plots for extant assemblages are, they necessarily monotonically increase towards the present and under a constant diversification rate, this increase is exponential. The integral therefore is necessarily weighted towards the number of lineages in recent evolutionary time compared to the number of lineages in deeper evolutionary time. In order to downweight the number of recent lineages when calculating TILD, we log-transformed the y-axis (i.e., the number of lineages at each point in time) prior to taking the integral.

*2.3. Statistical Analyses*

As the individual FIA plots are quite small in scale, we combined all plots within 0.2° grid cells prior to calculating ALD metrics (n = 13,207 grid cells). In order to determine the main evolutionary groups of tree assemblages across grid cells, we used k-means clustering of assemblages based on their shared phylogenetic branch length, as quantified by the Phylosorensen Index [33]. An elbow analysis suggested that 14 groups was a parsimonious number that minimised within group variation in phylogenetic composition (Figure S1). Preliminary analyses of the distribution of these groups over

geographic and climatic space showed that several pairs of groups overlapped both in geographic location and climatic environment. These pairs of groups were combined to give nine total evolutionary groups that were geographically and climatically cohesive. A silhouette analysis [34] was then run for these nine evolutionary groups in order to determine if any individual sites were closer in their evolutionary composition to the medoid value of another group than the group to which they were originally assigned (as measured by the Phylosorensen Index). If such was found, these sites were then reassigned to the group to which they were more similar in evolutionary composition.

In order to visualise the compositional relationships of these different groups, we ordinated assemblages based on the presence versus absence of evolutionary lineages, as quantified by the occurrence of individual nodes in the phylogeny in each assemblage. We specifically used the evolutionary principal component analysis developed by Pavoine (2016) [35], with the occurrence data Hellinger transformed prior to ordination [36]. This approach also allows identification of the evolutionary lineages that are associated with different components of the ordination space. We determined the lineages that are most strongly correlated with the first two principal components. In order to further characterise the composition of the evolutionary groups, we conducted a standard indicator analysis to determine the species most strongly associated with each group [37]. Lastly, to further characterise the evolutionary groups, we mapped where they occur in geographic and climatic space. In order to better visualise how the groups occupy geographic and climatic space, we generated 95% kernel density estimates [38] of the distribution of each group over two climatic dimensions, mean annual temperature and precipitation, and two geographic dimensions, elevation and latitude.

There is wide variation in the number of individual trees sampled across the combined plots in each grid cell ($887 \pm 1204$ inds, mean $\pm$ s.d.; range 2–17,307 inds), and all of the ALD metrics that we calculated, except sPD, were positively correlated with the number of individuals sampled (Pearson's $r = 0.60-0.76$). In order to obtain comparable estimates of ALD, we rarefied grid cells to the same number of individuals. While rarefaction can be problematic because it excludes assemblages from analysis below the abundance threshold used and introduces heteroscedasticity in the diversity estimate that is related to the number of individuals sampled [39], we do not know of any estimates of the richness dimension of ALD or phylogenetic diversity that are robust to variation in sample size (in terms of number of individuals sampled). While Rao's quadratic entropy has been proposed as an estimate of phylogenetic diversity that is robust to variation in sample size, it measures the divergence dimension of phylogenetic diversity, not the richness dimension [28], and was therefore not of interest to us here.

In order to determine the number of individuals to select in rarefactions, we first selected the subset of assemblages that have at least 1000 individuals (3660 grid cell assemblages). We estimated the species richness of these assemblages when rarefied to 1000 individuals (i.e., expected number of species per 1000 stems). We then rarefied these assemblages to smaller numbers of individuals, and observed how the richness estimate for a smaller number of stems correlated with the richness estimate per 1000 stems. Once assemblages were rarefied to less than 50 stems, the correlation (pearson's r) between the two richness estimates dropped below 0.95. We therefore chose 50 individuals as the size for rarefied assemblages. We repeated rarefactions 100 times, and calculated the average of each ALD metric over these 100 rarefactions.

In order to assess the general behaviour of ALD metrics, we calculated the spearman's rank correlation (rho) between a given ALD metric and the number of lineages at different phylogenetic depths (in intervals of 1 Myrs between the present and the root of the seed plant phylogeny at 350 Ma). We used spearman's rank correlation because these relationships are not necessarily linear, and because our goal is to evaluate if assemblages would be ranked similarly, e.g., for conservation prioritisation, if counting the number of lineages at a particular time slice vs. using a given ALD metric. In order to obtain an overall measure of the behaviour of a lineage diversity metric, we then obtained the mean of the spearman's rho values across all phylogenetic depths. All analyses were carried out in

the R Statistical Environment [22] using functions in the ape [40], picante [41], vegan [42], cluster [43], adiv [44] and PhyloMeasures [30] packages. The analysis script is available in Appendix A.

## 3. Results

Clustering analyses based on shared evolutionary history resulted in nine major evolutionary groups of assemblages, which vary in their geographic (Figure 3), elevational and climatic distributions (Figure 4). The west coast of the United States is dominated by a single group, but as one moves inland there are four different evolutionary groups that are spatially mixed across much of the western United States. They occupy relatively distinct regions of climatic space, and their spatial interdigitation likely results from environmental variation generated by topographic heterogeneity. In contrast, the four groups east of the Mississippi are clearly arranged in a latitudinal manner, reflecting the fact that environmental gradients are more gradual in the eastern United States (Figure 3). These groups clearly replace each other along a temperature gradient from colder to warmer sites (Figure 4B). There are two groups in the centre of the United States, one of which is most predominant in Texas, but also extends in a scattered distribution further north in the Great Plains and westwards into the southern Rockies. The other central group is scattered through the more eastern, wetter portions of the Great Plains and also in the Midwest, with its core extent in the northern Great Plains. More detailed comments regarding the groups can be found in Table S2.

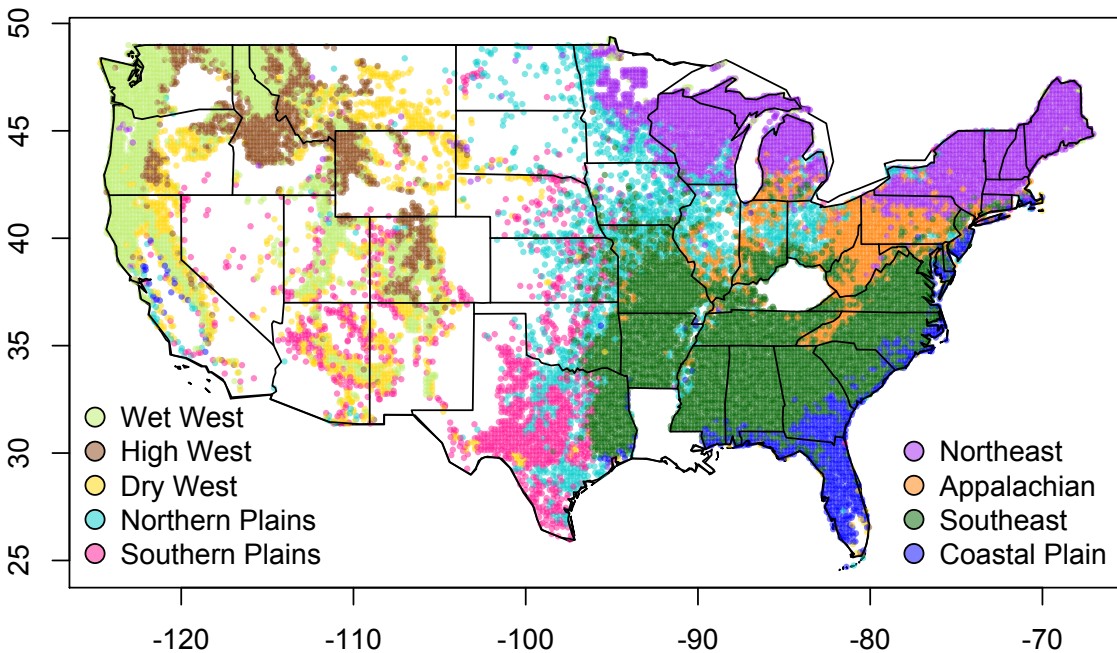

**Figure 3.** Map of tree assemblages included in this study, coloured by evolutionary group following a clustering analysis based on shared evolutionary history. Names for groups were chosen based on their geographic and/or climatic characteristics. Each assemblage consists of all FIA plots within a 0.2° by 0.2° grid cell.

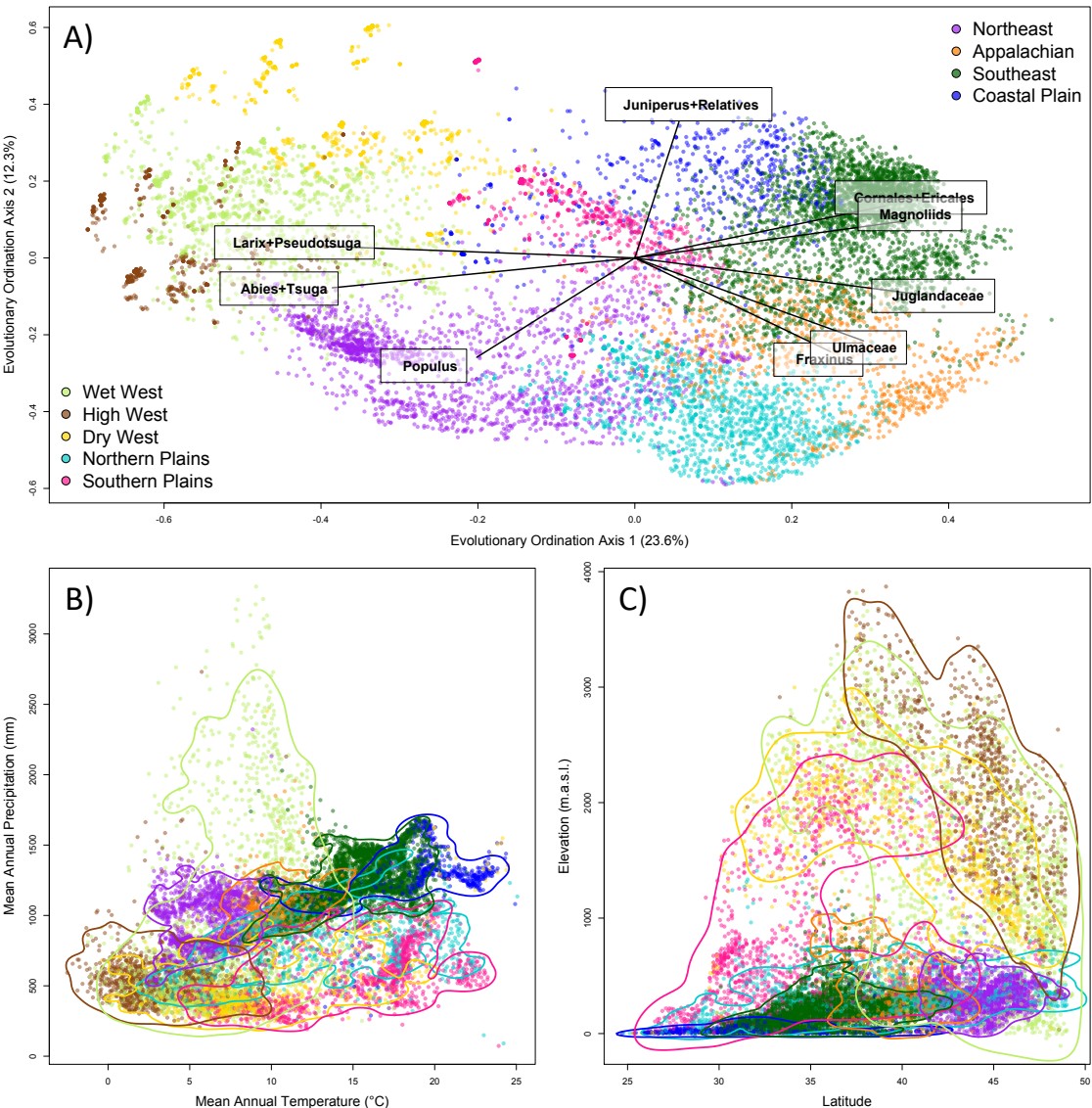

**Figure 4.** Distribution of nine main evolutionary groups of tree assemblages in the contiguous United States across: (**A**) an ordination space based on shared evolutionary history, with the influence of the key clades highlighted; (**B**) climatic space; and (**C**) elevation and latitude.

The majority of grid cell assemblages sampled at least 50 individuals (11,547 of 13,207 assemblages or 87%), and were therefore included in calculations of assemblage lineage diversity (ALD). We estimated ALD metrics for each assemblage, including constructing lineage through time (LTT) plots for each to calculate the time integrated lineage diversity (TILD). We show a sample of these LTT plots for each evolutionary group in Figure 5. There is clear variation across groups in when they accumulate lineage diversity. The Northern Plains group is composed entirely of eudicot angiosperms, and therefore most assemblages only have a single lineage (a log value of 0 on y-axis) until the eudicots begin to diversify ~120 Ma. An entirely contrasting pattern can be found in assemblages of the Dry West group, which have multiple lineages of gymnosperms, and thus have high lineage diversity deep in time. However, these assemblages are relatively poor in angiosperms and so do not achieve the same number of lineages in recent time periods as the Northern Plains group. The eastern groups have the highest number of lineages in recent time slices and also tend to have high lineage diversity deep in time, except for the Appalachian group which is relatively poor in gymnosperms.

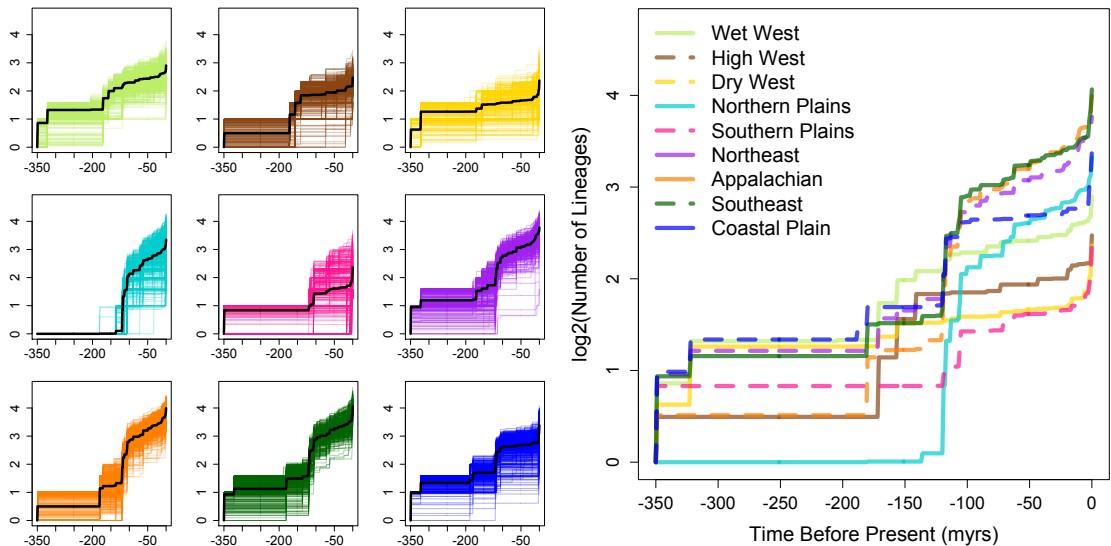

**Figure 5.** (**Left**) Lineage through time (LTT) plots for a sample of 500 assemblages from each evolutionary group, with each coloured line representing the average values over 100 rarefactions of the given assemblage to 50 individuals. Meanwhile, the thick black line gives the mean value at each evolutionary depth across all assemblages for a given evolutionary group. (**Right**) The mean number of lineages at each evolutionary depth across all assemblages in each group (thick black line from left panel), arrayed on one plot for direct comparison. Note log2 transformation of y-axes and that the number of lineages at the right-hand side of each graph represents the (mean) species richness of the assemblage(s).

Except for standardised phylogenetic diversity (sPD), the different ALD metrics we calculated are highly correlated with each other (Pearson's r = 0.60−0.95; Figure 6) and with the number of species in assemblages (Pearson's r = 0.57−0.93; Figure 6). The correlation of species richness (SR) with the number of lineages declines with increasing phylogenetic depth, dropping to very low values prior to the radiation of the Eudicots at 120 Ma (Figure 7). The other taxonomic measures of ALD all show a similar pattern to SR; i.e., none show a strong correlation with number of lineages prior to ∼120 Ma. Because of this, none of the taxonomic measures of lineage diversity show a mean correlation over all evolutionary depths greater than 0.5 (Figure 7).

The phylogenetically-derived metrics of ALD vary in their pattern of correlation with number of lineages over different evolutionary depths (Figure 7). Neither sPD or sum of evolutionary distinctiveness (sumED) show high mean correlations (mean rho = 0.30 and 0.46 respectively), and these two metrics show contrasting patterns over phylogenetic depth. sPD is more strongly correlated with the number of lineages deep in evolutionary time, while sumED shows a pattern more similar to taxonomic measures of lineage diversity. Time integrated lineage diversity (TILD) shows the highest mean correlation with number of lineages across all phylogenetic depths (mean rho = 0.76), but phylogenetic diversity (PD) and phylogenetic species richness (PSR) also showed relatively high mean correlations (mean rho = 0.67 and 0.66 respectively). PD and PSR show stronger correlations with the number of lineages in recent evolutionary time, while TILD shows stronger correlations with the number of lineages in deeper evolutionary time (Figure 7). Given that other taxonomic measures of ALD are strongly correlated with SR, that sPD and sumED show low mean correlations with number of lineages across most phylogenetic depths and that PSR does not show a different pattern from PD, with which it is highly correlated (r = 0.95; Figure 6), we focus below on patterns with respect to SR, PD and TILD.

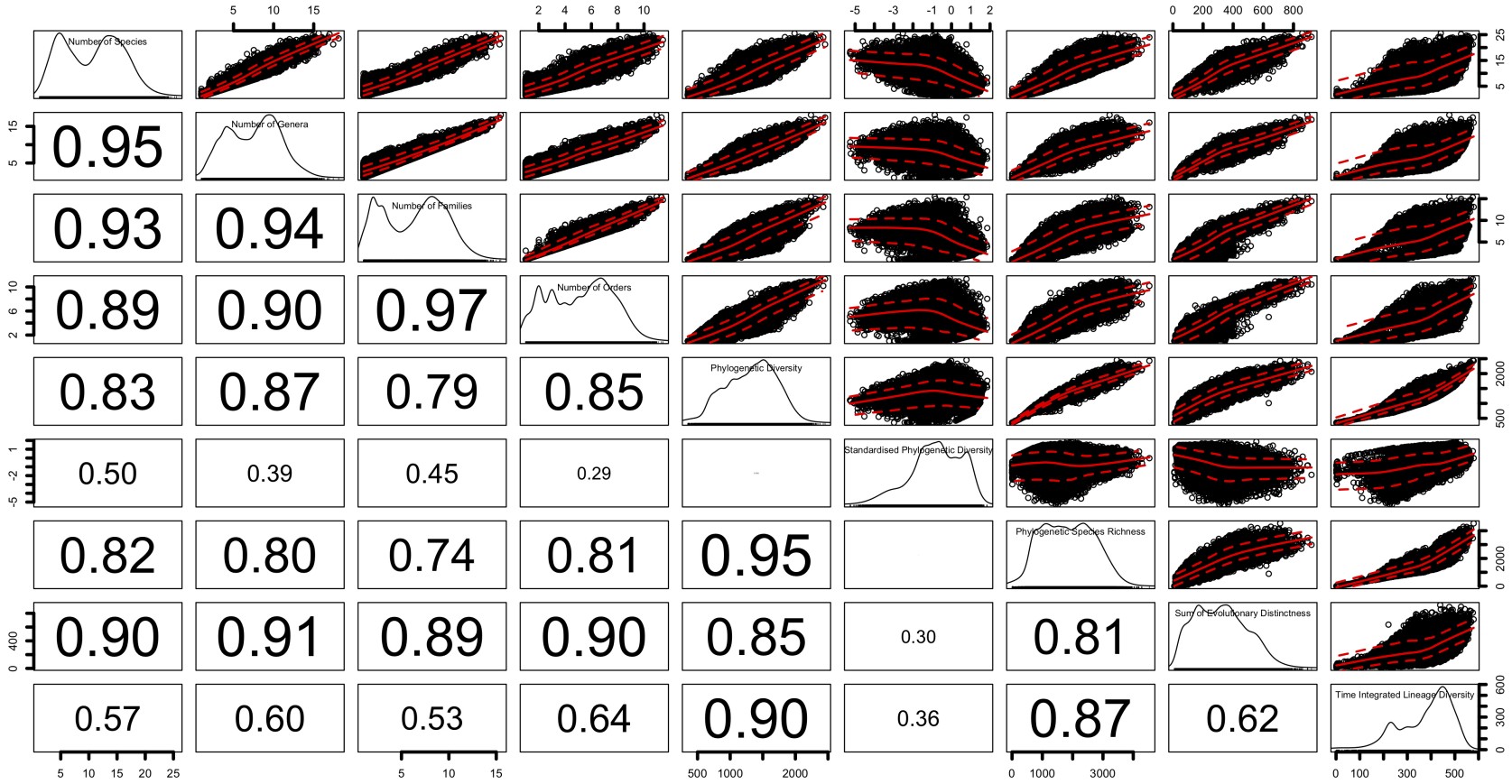

**Figure 6.** Pairwise relationships between lineage diversity metrics, with Pearson's correlation coefficient given on the lower half of the matrix. The solid red line represents a loess, moving average regression and the dashed red lines represent the conditional variability over the range of the x-axis. The diagonal gives probability density plots for each metric.

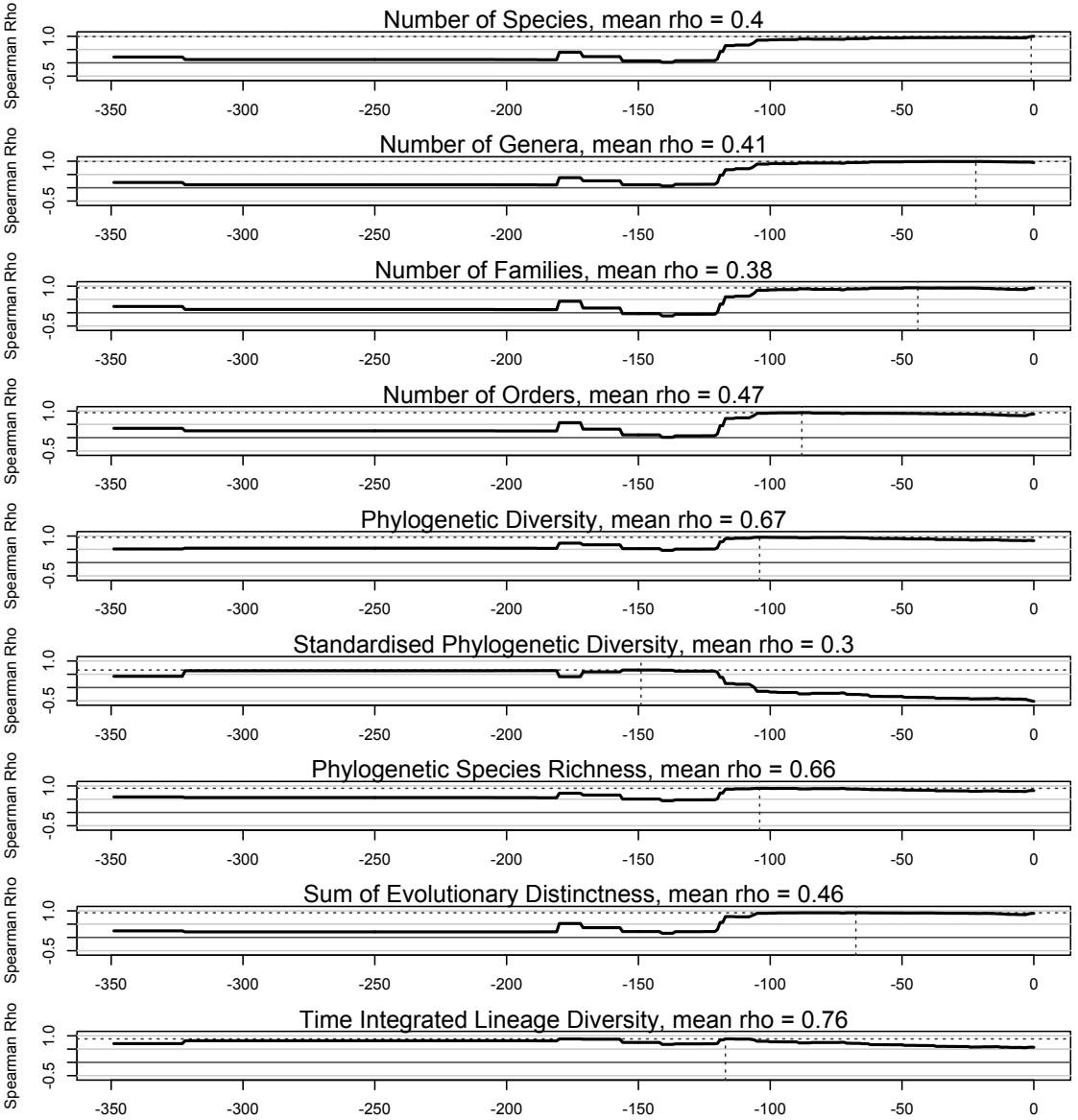

**Figure 7.** Spearman's rank correlations between different synthetic measures of assemblage lineage diversity and the number of lineages at different phylogenetic depths. The mean value of spearman's rho across all depths (excluding t = 0 and 350) is given above each plot. The phylogenetic depth at which the maximum correlation is found is marked with dashed lines going to the x and y-axes.

Species richness of assemblages, quantified as the number of species per 50 trees, shows clear spatial patterns across the contiguous United States (Figure 8). Low values are generally observed in the western half of the United States, while in the eastern half, low values are observed in Florida and parts of the Northeast. Among the western groups, the highest SR is found in assemblages in the Wet West group, while in the east, the highest values are in the Southeast group. PD shows similar patterns to SR (Figure 8), although it gives higher values on average for the Northeast group than the Appalachian group, while the opposite holds for SR. Also for PD, the Wet West group approaches values observed in the eastern groups, while that is not the case for SR. The TILD metric shows patterns that contrast with those for SR and PD (Figure 8). TILD gives higher values on average for the Southern Plains group than the Northern Plains group, while PD finds the opposite relative ranking. TILD also gives the Wet West group equal value to that for eastern groups. Within the east, TILD gives values for the Coastal Plain group equal to that for the Northeast and Southeast groups, while it gives relatively lower values for the Appalachian group. Overall, for most groups, PD shows a pattern for groups that

is intermediate between that observed for SR and TILD. In analysing the deviation of TILD from the expectation given PD (based on the residuals of a regression of TILD on the logarithm of PD), we see that TILD gives higher values than expected for the Wet West, Dry West, Southern Plains and the Coastal Plain, and lower values than expected for the Northern Plains and Appalachians (Figure 8, bottom row).

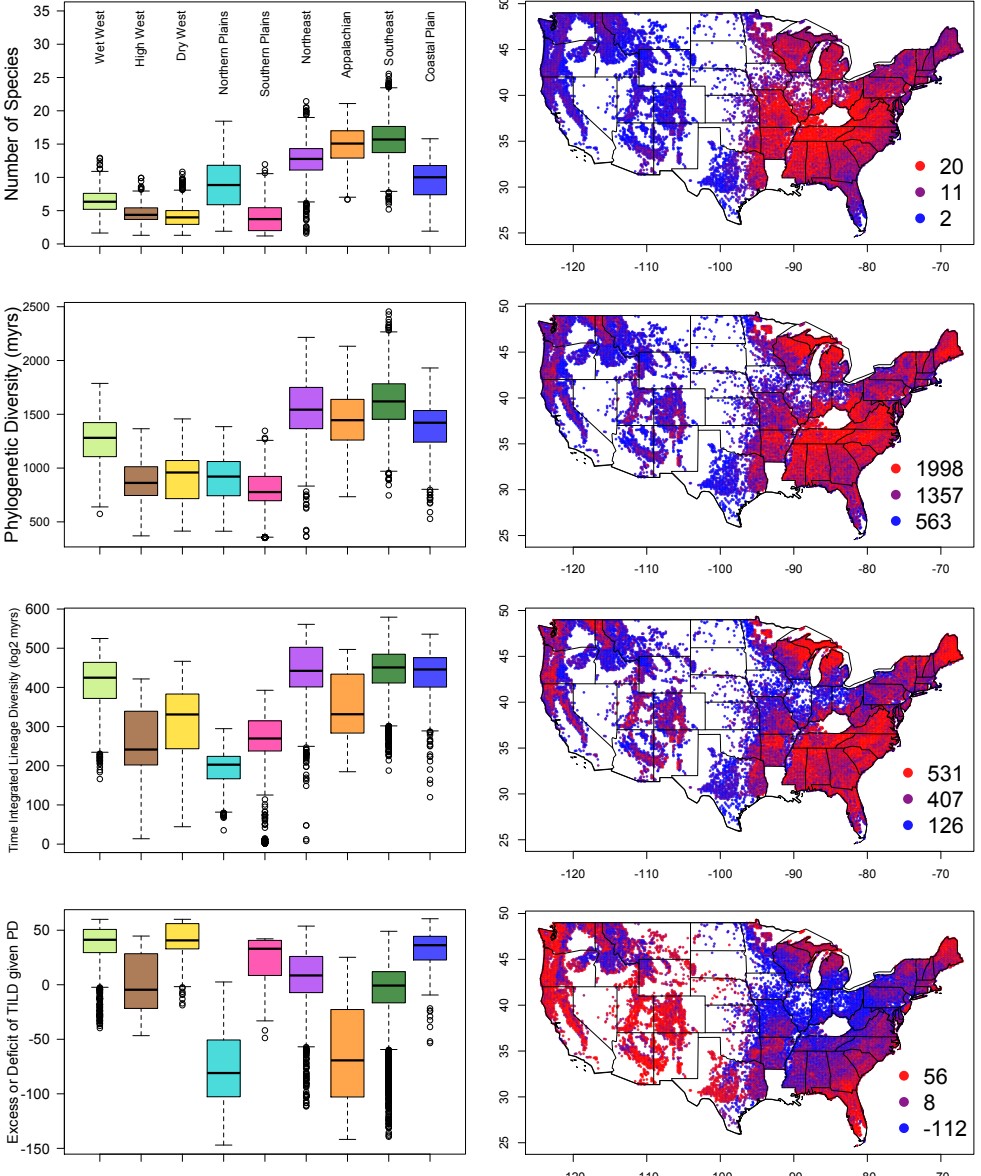

**Figure 8. Left:** Variation in species richness, phylogenetic diversity (PD), time integrated lineage diversity (TILD) and the excess or deficit of TILD given PD for tree assemblages, separated by major evolutionary group. The latter represent the residuals from a regression of TILD on the logarithm of PD. Values for each assemblage represent the average across 100 rarefactions to 50 tree individuals. **Right:** Maps of the same metrics, per assemblage, across the contiguous United States, with the colours for the upper 95% quantile, median and lower 95% quantile for a given metric in the lower right-hand corner of each map.

## 4. Discussion

Many different metrics could potentially be used to quantify the lineage diversity of organismal assemblages. For tree assemblages across the contiguous United States, we find that two metrics,

which can be derived from temporally calibrated phylogenies, show the greatest average correlation with number of lineages over the full evolutionary history of seed plants, and thus seem best suited to quantify assemblage lineage diversity (ALD). These are Faith's phylogenetic diversity (PD) and a metric that is newly derived here, time integrated lineage diversity (TILD). Other metrics derived from molecular phylogenies either showed lower average correlations with number of lineages in assemblages, or were strongly correlated with PD. Meanwhile, taxonomic metrics, including species richness (SR), failed to correlate with the number of lineages deep in evolutionary time, specifically, prior to the origin of the Eudicots. This is because the high number of species, genera, families and orders of Eudicots in assemblages in the eastern United States drive the pattern of variation in taxonomic metrics. If prioritisation schemes were to be based solely on SR, or other taxonomic richness measures of ALD, the entire western half of the US would receive less conservation attention than the eastern half. Yet, western US tree assemblages, dominated by older, relatively species-poor gymnosperm lineages, can still represent substantial reservoirs of evolutionary history, as reflected in TILD values comparable to the most lineage-diverse tree assemblages in the eastern US.

### 4.1. Taxonomic Measures of Lineage Diversity

In many studies [12,45], species richness has been found to be strongly correlated with phylogenetic diversity (PD), and has therefore been suggested as a suitable proxy for ALD [46]. Our study suggests that, at least for tree assemblages in the contiguous US, this is not the case. Higher-level taxonomic measures that we explored, specifically the numbers of genera, families and orders in assemblages, do not perform much better. As expected, as higher taxonomic ranks are used, strong correlations with number of lineages persist deeper into evolutionary time (compare the x-intercept of highest correlation for different taxonomic ranks in Figure 7), but none of the taxonomic measures provide a high correlation with number of lineages prior to ~120 Ma. This is perhaps unsurprising as the majority of lineages deeper in evolutionary time are gymnosperms (Figure 2), and all the gymnosperms in our dataset come from a single order, three families and 15 genera, while angiosperms dominate the variation in taxonomic measures of ALD with 18 orders, 35 families and 68 genera. Thus, for clades with highly imbalanced phylogenies, like seed plants, taxonomic measures of lineage diversity are not likely to provide an adequate, synthetic measure of ALD [47].

### 4.2. Phylogenetic Measures of Lineage Diversity

Time integrated lineage diversity (TILD) represents the area beneath a lineage through time plot where the number of lineages per time slice has been log-transformed. TILD is mathematically related to PD, which is identical to the area beneath a raw (i.e., non-log-transformed) lineage through time plot. PD was originally conceived as a metric to aid conservation prioritisation [19], and it has always been properly interpreted as a measure of the total evolutionary diversity in assemblages, which is certainly worth quantifying. But, it is strongly skewed towards the number of lineages present in recent evolutionary time, downweighting older evolutionary divergences. We suggest that researchers may use PD to quantify ALD more recently in evolutionary time, and complementarily, TILD may be more suitable to obtain a measure of ALD that gives equal weight to deeper evolutionary time as recent evolutionary time. While phylogenetic species richness (PSR; [29]) is strongly correlated with PD and could represent an alternative to it, we suggest researchers continue to use PD, because of its historical precedence and because, as with TILD, it is directly interpretable in terms of numbers of lineages.

For this dataset, the standardised phylogenetic diversity (sPD) correlates well with the number of lineages deep in evolutionary time, but not with numbers of lineages in recent evolutionary time. In fact, sPD is negatively correlated with numbers of lineages less than 70 million years old. We suggest that sPD may better serve as a metric of phylogenetic community structure, which is interesting in its own right [48], but that it should not be used as a measure of the richness dimension of lineage

diversity in assemblages. In contrast to sPD, the sum of evolutionary distinctiveness in assemblages, sumED, showed weaker correlations deeper in evolutionary time and stronger relationships in recent evolutionary time (Figure 7). In fact, sumED showed a very similar pattern to taxonomic metrics of lineage diversity (Figure 7), and of the phylogenetically-derived metrics in this study, it shows the strongest correlation with taxonomic metrics (Figure 6). As a conservation prioritisation metric, sumED has a clear intuitive value, since it represents the totality of phylogenetic diversity in a given assemblage that is rare in the entire dataset, but values for assemblages will be sensitive to phylogenetic taxon sampling in the overall dataset, even if a given assemblage itself is fully phylogenetically sampled (see Isaac et al., (2007) [32] for full explanation of how ED is derived for each species). Conversely, PD and TILD will only vary based on sampling within quantified assemblages, and are therefore more straightforward to apply.

### 4.3. Tree Diversity Patterns across the Contiguous United States

Consistent with previous assessments [49,50], the most evident spatial contrast in the species richness pattern is between the eastern and western United States. To a very coarse approximation, this reflects the dominance of gymnosperms in the western United States, the dominance of angiosperms in the eastern United States, and the fact that angiosperms are a much more diverse clade than gymnosperms (even when focusing only on trees). Previous studies have identified the high plateau south of the Appalachian Mountains [49], and the Florida panhandle, Alabama/Georgia border region [50], as areas of maximal tree species richness in the United States. In contrast we found the highest tree species richness in a region centred on Kentucky and Tennessee. This contrast in results could be due to the different spatial grain size of analysis, our use of plot data rather than overlap in range maps or the fact that we only include taxa larger than 12.7 cm dbh and thus exclude small tree taxa. The forest region centred on Kentucky and Tennessee corresponds to the mixed mesophytic forest region [51]. The first part of the name reflects that there are no particularly dominant tree species in the region and most forest stands have a mix of dominant species. Braun (1950) [51] recognised the exceptional tree species richness of this region and characterised it as "the association of the Deciduous Forest which occupies the area of optimum moisture and temperature conditions of North America" (p. 42). Indeed, moisture stress for plants is lower in this region of the US compared to regions southeast of the Appalachians or the entire western US, while temperatures do not reach the extreme lows that occur in the northern parts of the contiguous US. Meanwhile, as the second part of the name, mesophytic, implies, the forests in this area are also found on more fertile soils compared to other forests in the US. Thus, the high alpha diversity of tree assemblages in this region may reflect an environment that is the most benign for the majority of tree species occurring in the contiguous United States. This is similar to the pattern found in another large biogeographic region, the Amazon, where the most species-rich tree assemblages are found in the western Amazon, which has relatively fertile soils and is subject to less moisture stress than the southern or eastern Amazon [12].

The spatial patterns of PD and TILD for tree assemblages show several evident contrasts with the spatial pattern of tree species richness (SR). In general, the western US shows almost uniformly low values of SR, at least in comparison to the eastern US, but the PD and TILD metrics show much greater variation. In particular, the TILD metric shows values for assemblages in the Wet West group that are comparable to values for assemblages in the eastern United States. One evident hotspot of lineage diversity in the west is the temperate rain forest region of the Pacific northwest, with high PD and TILD values. This temperate rain forest region includes the 'Miracle Mile' in the Klamath mountains of northern California which holds 18 species of conifers [52], albeit not that many occur in any of the assemblages derived from FIA plots. PD and TILD plummet as one crosses eastward over the Cascade mountain range or the Sierra Nevada mountain range. Presumably the arid conditions on the eastern side of these mountain ranges limit tree lineage diversity. Though species richness minima to the east of the Sierra Nevada (as well as the Rocky Mountains) have previously been noted [49], this contrast across mountain ranges is most evident when considering PD, and particularly TILD. Other regions

of notable ALD in the western US include scattered areas in the southern Rocky Mountains and northern Idaho.

In the eastern United States, the spatial pattern of PD, and particularly TILD, also contrasts with that of species richness, although there are areas that are high for all three of these diversity measures (e.g., much of the Southeast). The most north-eastern state in the contiguous US (Maine) as well as the area around the Great Lakes emerge as regions of high PD and TILD, which is presumably due to the increasing prevalence of conifers in the far north and their deep evolutionary heritage. Meanwhile, the mixed mesophytic forest region that has the highest tree species richness values does not show the highest PD and TILD values. Higher PD and TILD are found to the south and east of the mixed mesophytic forest region. The south-eastern United States was highlighted as a region of high angiosperm tree PD in a previous study based on range maps [23], and our results show this is consistent when using inventory data, and when incorporating gymnosperms into the quantification of overall seed plant PD.

The Coastal Plain and Southern Plains groups are both notable in showing substantially higher TILD than SR values relative to other groups, which may be due to the incursion of tropical angiosperm lineages into these southern areas. For example, the Coastal Plain group is home to *Sabal* species [Arecaeae], *Persea borbonia* [Lauraceae], *Annona glabra* [Annonaceae], among other species, which belong to old, largely tropical families. Indeed, the average age of angiosperm families in the coastal plain is higher than anywhere else in the contiguous US [53]. Meanwhile, the Southern Plains group has among the lowest values for SR, but shows intermediate values for TILD, and includes represents of tropical dry region genera such as *Prosopis* and *Vachellia* (Fabaceae), which do not occur elsewhere in the contiguous United States.

## 5. Conclusions

Our study has explored the concept of lineage diversity, and how it might be quantified, for tree assemblages across the contiguous United States. We have shown how temporally calibrated molecular phylogenies can be used to quantify assemblage lineage diversity (ALD) and aid conservation prioritisation [19]. As might be expected, metrics derived from a molecular phylogeny showed stronger correlations than taxonomic metrics with the numbers of lineages over the evolutionary history of the focal clade (seed plants). As for specific recommended metrics, we suggest that phylogenetic diversity (PD) and time integrated lineage diversity (TILD) metrics both be used. PD has precedence in the literature and is useful for comparison with previous studies. We do stress however that PD is skewed towards the number of lineages in recent evolutionary time, while the newly-derived TILD metric is shown to better represent the entire evolutionary history of the clade of interest, and should therefore also be used.

We employed an empirical dataset on tree assemblages of the contiguous United States to explore these different metrics of ALD. We found that the spatial patterns of PD and TILD differ in important ways from the spatial patterns of species richness, for example by highlighting the high conservation value of temperate rainforests in the Pacific Northwest. PD and TILD also give somewhat contrasting results, with the former giving relatively higher values for tree assemblages in the northern Great Plains, Midwest and high elevation areas of the Appalachians, and the latter giving relatively higher values for some dry areas in the western US and the southeast Coastal Plain. However, it would be naïve to suggest that conservationists in the United States are unaware of the high conservation value of these different forests. Indeed, the tree flora of the United States is likely well enough known, such that good awareness already exists regarding which areas have particularly high or low conservation value with respect to tree species composition and lineage diversity. Where these metrics may be particularly useful is in less well known floras, such as in many tropical biogeographic regions. There has been one study of variation in lineage diversity across ~300 sites in the rain forests of the Amazon basin [12], but we know of no such similar study outside of tropical moist forest, or in other tropical regions.

**Supplementary Materials:** The following are available online at http://www.mdpi.com/1999-4907/10/6/520/s1, Table S1: Species added to the molecular phylogeny, Table S2: Descriptions of the major evolutionary groups in the tree flora of the contiguous United States, Figure S1: Clustering validation (Elbow plot).

**Author Contributions:** K.G.D. and R.A.S. conceived the manuscript and led analyses. A.R.G. contributed to analyses. All authors contributed to writing.

**Funding:** This research received no external funding.

**Acknowledgments:** We thank two anonymous reviewers for their thorough and insightful reviews, which greatly improved the quality of the manuscript. K.G.D. was supported by a Leverhulme International Academic Fellowship during the time this research was completed. R.A.S. is supported by Newton International Fellowship from The Royal Society, Conicyt PFCHA/Postdoctorado Becas Chile/2017 N° 3140189 and CONICYT PIA APOYO CCTE AFB170008. A.R.G. is supported by a NERC studentship (NE/L002558/1).

**Conflicts of Interest:** The authors have no conflicts of interest to report.

## Appendix A

R Codes to execute analyses.

## Appendix B

Combined molecular phylogeny derived from combining the angiosperm and gymnosperm phylogenies in Ma et al., 2016, with the addition of unsampled species.

## Appendix C

Taxonomy table used to determine taxonomic richness of tree assemblages.

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
