# Peer review of "Exploring the Concept of Lineage Diversity across North American Forests"

_forests, doi:10.3390/f10060520_

Round 1

Reviewer 1 Report

The manuscript “Exploring the Concept of Lineage Diversity across North American Forest” recovers the main objectives of the previous version (Comparing Measures of Community Lineage Diversity across North American Forests) improving and / or eliminating those aspects of the original that could be more controversial. In addition, the authors propose a new analysis of the original data, which I think is fully accurate and useful for the reader, not only in its approach but in its interpretation also. Using their words “To give context to our results, we conduct a clustering analysis of assemblages based on their shared evolutionary history, thereby determining the main evolutionary groups of tree assemblages in the contiguous United States”.

I would like to acknowledge the great effort made by the authors adapting the manuscript to the comments of the reviewers. It has not been only about changing "words" (eg, assemblage instead community) but about rethinking the grouping of the original data, and reanalyzing the results for testing again their initial hypotheses. In this sense, and always from my point of view, it has been a good job.

Finally, I will add that new manuscript is not easy to read relative to the description, presentation of results and comparison of the different indexes, but I cannot see an alternative in  order to arguing  that the new index proposed (time integrated lineage diversity, or TILD) provides a new dimension to incorporate the full evolutionary time in the assessment of the phylogenetic diversity of a set of species, whether it is an assemblage of species, a community, an ecosystem or a regional biota.

My highlight from the manuscript could be “…… the case of North American trees gives value to the lineages of Gymnosperms, with an actual lower taxonomic diversification than Angiosperms but with a much longer evolutionary history, which undoubtedly provides a valuable insight for their conservation”.

Author Response

We thank the reviewer for their favourable review of this revision. We apologise that it is not the easiest manuscript to read, and we will hope this does not affect our citation rate!

p { margin-bottom: 0.25cm; direction: ltr; color: rgb(0, 0, 0); line-height: 115%; text-align: left; }p.western { font-family: "Calibri", serif; font-size: 12pt; }p.cjk { font-family: "Calibri"; font-size: 12pt; }p.ctl { font-size: 12pt; }

Reviewer 2 Report

I am glad that the authors have taken my previous comments and criticisms seriously. I think the manuscript (ms) is greatly improved as a result.  The new analyses, change of grain size, clusters-based approach, etc, are all improvements in my opinion, and produce some nice results that mean that I agree with the authors that this is now better suited to being a research paper than a methods paper.  Indeed, the conclusion described at the end of the first paragraph of the Discussion strikes me as quite a highly citable empirical finding.

As outlined below, I think some tidying up is still needed, particularly of the main messages, but the research can be a useful contribution to the literature.

While the abstract has been rewritten well, I think the main messages need straightening out a bit in other parts of the ms, in several respects:

1. ‘Performance’ of the metrics.  It is good that the authors have recognised the circularity problem and made changes accordingly.  But a few vestiges of the previous approach remain.  Here are the examples I noted, but the authors should read through carefully to check there are no more:

*L40: ‘we test various metrics’. This implies that there is an evaluation of which metric is ‘better’, but as before the only yardstick here is comparison with results that are basically just empirical estimates of TILD for sampled time-slices.  Need to change ‘test’ to a more appropriate word - ‘explore’ is used elsewhere in the ms and would seem to be fine here.

*L209-11: ‘In order to obtain an overall measure of the ‘performance’ of a lineage diversity metric, we then obtained the mean of the spearman’s rho values at all phylogenetic depths.’ Just putting the word performance in quotation marks (the only change to this sentence from the original version of the ms) does not solve the circularity problem. Again, language more appropriate to the new approach is needed - language that is about exploring the behaviour of the metric, rather than ~‘how well it performs’.

L 286-7: [TILD and PD] ‘best capture the concept of assemblage lineage diversity (ALD).’  Again, this is misleading because it implies that there is an objective way of testing this. These metrics best match the ~empirical estimate of TILD.  It should be rephrased appropriately.

L314-5: ‘Time integrated lineage diversity (TILD) shows the highest average correlation over all evolutionary depths with number of lineages in assemblages.’  While I don’t think there is anything wrong with this statement per se, the phrasing combined with the fact that it opens a section of the discussion is potentially misleading - implying that it somehow performs better.

L403: ‘performed better than’. Inappropriate, as above.

L406-7: ‘This gives support to the notion that molecular phylogenies have a key role to play in aiding conservation prioritisation [19].’ I fail to see the basis for this ‘support’. The empirical work referred to shows that TILD and PD measure particular aspects of molecular phylogenies. It tells us nothing about whether molecular phylogenies aid conservation prioritisation - that seems to be your philosophical starting-point, rather than your empirical finding.

2. Is TILD a new measure to use on its own, or as a complementary measure?  In most of the ms it is put forward as a complementary measure (to PD), which I agree with.  For example, in the Abstract and L407 in the Conclusions section.  However, L68 says ‘It would be ideal to have a single, synthetic metric for ALD that integrates over the evolutionary history of the clade being studied’.  This seems to be a relict of the original ms (with its problematic emphasis on testing) that does not belong in the new version.

L323: TILD ‘focuses on deeper evolutionary history’ than PD. While it is strictly true (TILD does give more weight to deeper history than PD does), this strikes me as being a bit confusing because (if I have understood correctly) the point of logging is to give EQUAL weight across evolutionary history. Indeed, this is what the Abstract suggests. (This logging is the difference between TILD and PD, and is based on a hypothesis of an exponential function.)

My other main comments concern the supplementary material:

First, I think both Figs S2 and S3 should be in the main text.  This would fit the greater emphasis on comparing metrics rather than evaluating them.  I also note that both figures are discussed quite a lot in the ms (especially Fig. S3).

Second, L430-6 talk about Appendices A-C. These are different from the Supplementary materials, which is confusing.  Importantly, I did not find any of Appendices A-C in the materials provided to me for review.

SPECIFIC COMMENTS

L177-9: ‘A silhouette analysis was then run for these nine evolutionary groups in order to reassign potentially mis-assigned grid cell assemblages to the group to which they are most similar in evolutionary composition [34].’  This needs explaining. Few people will know what a silhouette analysis is, let alone how it works. Also, I wonder whether anyone could repeat your analysis from this description (I am not familiar with this analysis, so do not know). Perhaps the R code would help, but I did not find it (see comments above about missing Appendices).

L180-2: ‘In order to characterise these groups, we used indicator analyses to determine the species strongly associated with each group, conducted ordination analyses based on shared phylogenetic history [35], and mapped where these groups occurred in geographic and climatic space.’ Again, this needs explaining so that readers have a better idea of what was done, here, without having to read reference 35, and to enable someone to repeat your analyses.

L200: should this say ‘per 1000 stems’ rather than ‘per 500 stems’?

Figure 4: need to explain what the envelopes (shapes) are.  Some of the points fall outside all the envelopes, indicating that you have not simply drawn a line around all points in each group.  Presumably the envelopes are based on some function that excludes the X% of points that are furthest from the rest, or something like that.

Figure 4: it is difficult to figure out which colour is which. Please label the envelopes in B and C.  In A, please similarly indicate which cluster of points is which, probably by putting a label on the main mass of points in each group.

Figure 5: not clear.  Many non-integer numbers of lineages appear at all times in Figure 5. I guess this is because for each of the 500 assemblages shown an average of 100 rarefied samples is plotted. Whatever the reason, it needs to be explained in the caption (as is the case for Figure 6).

L250-263: many times correlations are given using r, which would be Pearson’s r, but according to Figure S3 these should all say rho (Spearman’s).  (Paragraph starting ‘The phylogenetically-derived metrics of ALD vary in their pattern of correlation’.)

L349-53: ‘Previous studies have identified the high plateau south of the Appalachian Mountains [46], and the Florida panhandle, Alabama/Georgia border region [47], as areas of maximal tree species richness in the United States. In contrast we found the highest tree species richness in a region centred on Kentucky and Tennessee, which corresponds to the mixed mesophytic forest region [48].’ This implies a contradiction, and that perhaps the previous studies were wrong.  My guess is that these differences reflect differences in grain size and/or the use of plot data vs range map overlays.  I think it is worth stating this, or similar.

L413-4: ‘We found that the spatial patterns of PD and TILD differ in important ways from the spatial patterns of species richness, for example by highlighting the high conservation value of temperate rainforests in the Pacific Northwest.’ Fine, but you are introducing a new metric (TILD), and this Conclusions section says nothing about how it differs from the existing PD. Surely a sentence on how the spatial patterns of TILD differ from those of PD should be added to the Conclusions.

The names of the variables in Figure S2 are too small.

Figure S3 seems to have a problem.  I cannot understand what the vertical dotted lines show. The caption says they show the ‘phylogenetic depth at which the maximum correlation is found’, but this does not appear to be true.  Please fix or explain.

MINOR

‘Is comprised of’ is incorrect.  I think a lot of people confuse ‘comprises’ with ‘is composed of’ or ‘consists of’, all of which mean much the same thing.  This needs correcting in various places (e.g. L145, L234)

L147: surely ‘derived from’, not ‘derived of’?!

Author Response

I am glad that the authors have taken my previous comments and criticisms seriously. I think the manuscript (ms) is greatly improved as a result.  The new analyses, change of grain size, clusters-based approach, etc, are all improvements in my opinion, and produce some nice results that mean that I agree with the authors that this is now better suited to being a research paper than a methods paper.  Indeed, the conclusion described at the end of the first paragraph of the Discussion strikes me as quite a highly citable empirical finding.

>> We thank the reviewer for their kind overall assessment and we also hope the main empirical finding is highly citable!

As outlined below, I think some tidying up is still needed, particularly of the main messages, but the research can be a useful contribution to the literature.

While the abstract has been rewritten well, I think the main messages need straightening out a bit in other parts of the ms, in several respects:

1. ‘Performance’ of the metrics.  It is good that the authors have recognised the circularity problem and made changes accordingly.  But a few vestiges of the previous approach remain.  Here are the examples I noted, but the authors should read through carefully to check there are no more:

>> Check for other instances!!

*L40: ‘we test various metrics’. This implies that there is an evaluation of which metric is ‘better’, but as before the only yardstick here is comparison with results that are basically just empirical estimates of TILD for sampled time-slices.  Need to change ‘test’ to a more appropriate word - ‘explore’ is used elsewhere in the ms and would seem to be fine here.

>> Changed ‘test’ to ‘explore’.

*L209-11: ‘In order to obtain an overall measure of the ‘performance’ of a lineage diversity metric, we then obtained the mean of the spearman’s rho values at all phylogenetic depths.’ Just putting the word performance in quotation marks (the only change to this sentence from the original version of the ms) does not solve the circularity problem. Again, language more appropriate to the new approach is needed - language that is about exploring the behaviour of the metric, rather than ~‘how well it performs’.

>> Changed ‘performance’ to ‘behaviour’.

L 286-7: [TILD and PD] ‘best capture the concept of assemblage lineage diversity (ALD).’  Again, this is misleading because it implies that there is an objective way of testing this. These metrics best match the ~empirical estimate of TILD.  It should be rephrased appropriately.

>> Sentence changed to ‘For tree assemblages across the contiguous United States, we find that two metrics, which can be derived from temporally calibrated phylogenies, show the greatest average correlation with number of lineages over the full evolutionary history of seed plants, and thus seem best suited to quantify assemblage lineage diversity (ALD).’

L314-5: ‘Time integrated lineage diversity (TILD) shows the highest average correlation over all evolutionary depths with number of lineages in assemblages.’  While I don’t think there is anything wrong with this statement per se, the phrasing combined with the fact that it opens a section of the discussion is potentially misleading - implying that it somehow performs better.

>> Sentence deleted.

L403: ‘performed better than’. Inappropriate, as above.

>> Please see next response.

L406-7: ‘This gives support to the notion that molecular phylogenies have a key role to play in aiding conservation prioritisation [19].’ I fail to see the basis for this ‘support’. The empirical work referred to shows that TILD and PD measure particular aspects of molecular phylogenies. It tells us nothing about whether molecular phylogenies aid conservation prioritisation - that seems to be your philosophical starting-point, rather than your empirical finding.

>> We have rephrased these two sentences as, “We have shown how temporally calibrated molecular phylogenies can be used to quantify assemblage lineage diversity (ALD) and aid conservation prioritisation \cite{faith1992conservation}. As might be expected, metrics derived from a molecular phylogeny showed stronger correlations than taxonomic metrics with the numbers of lineages over the evolutionary history of the focal clade (seed plants).”

2. Is TILD a new measure to use on its own, or as a complementary measure?  In most of the ms it is put forward as a complementary measure (to PD), which I agree with.  For example, in the Abstract and L407 in the Conclusions section.  However, L68 says ‘It would be ideal to have a single, synthetic metric for ALD that integrates over the evolutionary history of the clade being studied’.  This seems to be a relict of the original ms (with its problematic emphasis on testing) that does not belong in the new version.

>> Changed to, “It would be ideal to have metrics for ALD that integrate over the evolutionary history of the clade being studied.”

L323: TILD ‘focuses on deeper evolutionary history’ than PD. While it is strictly true (TILD does give more weight to deeper history than PD does), this strikes me as being a bit confusing because (if I have understood correctly) the point of logging is to give EQUAL weight across evolutionary history. Indeed, this is what the Abstract suggests. (This logging is the difference between TILD and PD, and is based on a hypothesis of an exponential function.)

>> Changed to, “TILD may be more suitable to obtain a measure of ALD that gives equal weight to deeper evolutionary time as recent evolutionary time.”

My other main comments concern the supplementary material:

First, I think both Figs S2 and S3 should be in the main text.  This would fit the greater emphasis on comparing metrics rather than evaluating them.  I also note that both figures are discussed quite a lot in the ms (especially Fig. S3).

>> We have now put these figures in the main text as Figs 6 and 7.

Second, L430-6 talk about Appendices A-C. These are different from the Supplementary materials, which is confusing.  Importantly, I did not find any of Appendices A-C in the materials provided to me for review.

>> Now that analyses are finalised for the manuscript, the R script will be cleaned up in order to make it available with the main text. Our instinct was to make it a separate text file to make it easier for people to work with, rather than including it as a part of the pdf supplementary material. Similarly, our instinct was to make the phylogeny and taxonomy table available as appendix files as they are not in pdf format (one is newick text format, other is csv file), as that is the format with which they are imported in the R script. If the journal/editors want things formatted differently, please just let us know.

SPECIFIC COMMENTS

L177-9: ‘A silhouette analysis was then run for these nine evolutionary groups in order to reassign potentially mis-assigned grid cell assemblages to the group to which they are most similar in evolutionary composition [34].’  This needs explaining. Few people will know what a silhouette analysis is, let alone how it works. Also, I wonder whether anyone could repeat your analysis from this description (I am not familiar with this analysis, so do not know). Perhaps the R code would help, but I did not find it (see comments above about missing Appendices).

>> Indeed, R code will be supplied with the article once accepted, such that analysis can be repeated. The authors do not have time to do a full clean up prior to submitting this revision, given the 3 day turnaround deadline (!!). The reassignment using silhouette analysis is a straightforward procedure, which we now try to describe better in the main text as follows, “A silhouette analysis [Rousseeuw 1987] was then run for these nine evolutionary groups in order to determine if any individual sites were closer in their evolutionary composition to the medoid value of another group than the group to which they were originally assigned (as measured by the Phylosorensen Index). If such was found, these sites were then reassigned to the group to which they were more similar in evolutionary composition.”

L180-2: ‘In order to characterise these groups, we used indicator analyses to determine the species strongly associated with each group, conducted ordination analyses based on shared phylogenetic history [35], and mapped where these groups occurred in geographic and climatic space.’ Again, this needs explaining so that readers have a better idea of what was done, here, without having to read reference 35, and to enable someone to repeat your analyses.

>> As these analyses are not the focus of our paper, we do not wish to spend excessive space on explaining them. But, we have expanded the paragraph as follows, “In order to visualise the compositional relationships of these different groups, we ordinated assemblages based on the presence versus absence of evolutionary lineages, as quantified by the occurrence of individual nodes in the phylogeny in each assemblage. We specifically used the evolutionary principal component analysis developed by Pavoine [2016], with the occurrence data Hellinger transformed prior to ordination [Legendre & Gallagher 2001]. This approach also allows identification of the evolutionary lineages that are associated with different components of the ordination space. We determined the lineages that are most strongly correlated with the first two principal components. In order to further characterise the composition of the evolutionary groups, we conducted a standard indicator analysis to determine the species most strongly associated with each group [Dufrene & Legendre 1997]. Lastly, to further characterise the evolutionary groups, we mapped where they occur in geographic and climatic space.”

L200: should this say ‘per 1000 stems’ rather than ‘per 500 stems’?

>> Fixed.

Figure 4: need to explain what the envelopes (shapes) are.  Some of the points fall outside all the envelopes, indicating that you have not simply drawn a line around all points in each group.  Presumably the envelopes are based on some function that excludes the X% of points that are furthest from the rest, or something like that.

>> We apologise for not explaining this sufficiently well. These are 95% kernel density estimates. We now explain that in the main text, “In order to better visualise how the groups occupy geographic and climatic space, we generated 95% kernel density estimates [Duong 2007] of the distribution of each group over two climatic dimensions, mean annual temperature and precipitation, and two geographic dimensions, elevation and latitude.”

Figure 4: it is difficult to figure out which colour is which. Please label the envelopes in B and C.  In A, please similarly indicate which cluster of points is which, probably by putting a label on the main mass of points in each group.

>> With regard to panel A, we feel that additional labelling within the figure would overly complicate it, given that there are already labels for the major lineages in the figure. We have discussed with a few colleagues and they seem to find the legends to be sufficient to identify groups and distinguish them. Similarly, as many of the groups overlap spatially in B and C, it is not clear how to put a label in the middle of the mass of points without the labels overlapping. We included the 95% kernel density estimates to try and make the patterns more clear, and based on discussions with colleagues, they could this approach useful.

Figure 5: not clear. Many non-integer numbers of lineages appear at all times in Figure 5. I guess this is because for each of the 500 assemblages shown an average of 100 rarefied samples is plotted. Whatever the reason, it needs to be explained in the caption (as is the case for Figure 6).

>> Indeed, the reviewer has divined the reason why these individual lines do not seem to accumulate lineages as whole numbers. This is because it is an average over 100 rarefied communities. We now include this information in the figure legend.

L250-263: many times correlations are given using r, which would be Pearson’s r, but according to Figure S3 these should all say rho (Spearman’s). (Paragraph starting ‘The phylogenetically-derived metrics of ALD vary in their pattern of correlation’.)

>> Thank you for catching this. Now fixed.

L349-53: ‘Previous studies have identified the high plateau south of the Appalachian Mountains [46], and the Florida panhandle, Alabama/Georgia border region [47], as areas of maximal tree species richness in the United States. In contrast we found the highest tree species richness in a region centred on Kentucky and Tennessee, which corresponds to the mixed mesophytic forest region [48].’ This implies a contradiction, and that perhaps the previous studies were wrong.  My guess is that these differences reflect differences in grain size and/or the use of plot data vs range map overlays.  I think it is worth stating this, or similar.

>> We have now included some thoughts on why we obtained different results, “This contrast in results could be due to the different spatial grain size of analysis, our use of plot data rather than overlap in range maps or the fact that we only include taxa larger than 12.7 cm dbh and thus exclude small tree taxa.”

L413-4: ‘We found that the spatial patterns of PD and TILD differ in important ways from the spatial patterns of species richness, for example by highlighting the high conservation value of temperate rainforests in the Pacific Northwest.’ Fine, but you are introducing a new metric (TILD), and this Conclusions section says nothing about how it differs from the existing PD. Surely a sentence on how the spatial patterns of TILD differ from those of PD should be added to the Conclusions.

>> We have now included this sentence, “PD and TILD also give somewhat contrasting results, with the former giving relatively higher values for tree assemblages in the northern Great Plains, Midwest and high elevation areas of the Appalachians, and the latter giving relatively higher values some dry areas in the western US and the southeast Coastal Plain.”

The names of the variables in Figure S2 are too small.

>> We have now served up this figure in landscape format, which will allow it to be larger and for the variable names have a larger font size.

Figure S3 seems to have a problem.  I cannot understand what the vertical dotted lines show. The caption says they show the ‘phylogenetic depth at which the maximum correlation is found’, but this does not appear to be true.  Please fix or explain.

>> Many thanks for catching this. There was a bug in our code. This has now been fixed.

MINOR

‘Is comprised of’ is incorrect.  I think a lot of people confuse ‘comprises’ with ‘is composed of’ or ‘consists of’, all of which mean much the same thing.  This needs correcting in various places (e.g. L145, L234)

>> Comprised changed to composed in these instances.

L147: surely ‘derived from’, not ‘derived of’?!

>> Changed to ‘composed of’.

p { margin-bottom: 0.25cm; line-height: 115%; }

This manuscript is a resubmission of an earlier submission. The following is a list of the peer review reports and author responses from that submission.

Round 1

Reviewer 1 Report

The manuscript “Comparing Measures of Community Lineage Diversity across North American Forests” raises, both in the title and in the main objective, the comparison of different indices (including a new one proposed by the authors) that measure the community lineage diversity (CLD). For this purpose the authors use a very large database (Forest Inventory and Analysis -FIA- Program of the US Forest Service) that extend over 177.000 plots, and the corresponding phylogeny of all tree species based on the phylogenies for gymnosperms and angiosperms in Ma et al. (2016). Although the authors describe in the Material and Methods section (2.3 Statistical Analyses) a method of rarefaction to make comparable the estimates of CLD in each of the plots, resulting in a final number around 125,000 plots, the systematic use of the term community to designate them seems confusing to me (and also inappropriate) from a forest ecology point of view. I think that assemblage (or species assemblage) as used for example in the aforementioned Tucker et al, 2017, is it much more accurate because each plot is not itself a tree community or forest community. This commentary makes sense because all measurements are made at the level of local diversity (α diversity) and in the dimension 0H (following the nomenclature of Hill numbers) calculated basically as species richness. Only in the case of aggregation of plots according to their similarity in the composition of tree species should we talk about of communities and / or ecosystems and calculate their corresponding CLD metrics as ϒ diversity. Following this thread a question arises, what is the value of these indices, and at this scale, in order to establish conservation strategies? Currently, the use of species richness as a surrogate to define units or conservation areas is far exceeded. The presence of endemic species at different levels, the genetic diversity of the populations of the different species, their adaptive capacity to current or future conditions (not to mention their sustainable use) are insight to be taken into account together with the CLD measures that you propose. I mean with this that the argument that these metrics improve the possibility of establishing conservation priorities in relation to the metric of species richness does not have much weight.

On the other hand, and in view of the results obtained, the authors draw Figure 5 with its legend “Spatial pattern of species richness and phylogenetically-derived community lineage diversity metrics per 25 trees for individual plots across the contiguous United States (n = 124,566 plots), with plots coloured according to their value. The legends gives the maximum and minimum values”. As far as I can see using zoom + there are only points of two colors, red and blue, which correspond to maximum and minimum values in the legend but nothing is said about how each point is assigned as low or high value. Without being this the most important thing, when the authors remark the spatial pattern of several indices and when they compare between them (section 4.3. Tree diversity patterns across the contiguous United States), are they based in a visual comparison about the colors of the map? I sincerely believe that speaking about spatial pattern, and not about apparent distribution or gross distribution, involves the use of tools based on geographic information systems that can detect not only the spatial patterns at different scales but also the spatial relationship derived from the divergence of the indices (species richness versus CLD).

Another question that arises is related to the construction of Lineage through Time (LTT) curves. I have seemed to understand that the intervals of time to calculate each slice are 5 Ma, which means that in constructing the whole of the curve, approximately 70 points are calculated. Since Figure 4 calculates the correlation of the values of the different indices with the number of lineages at different phylogenetic depths, I would like to know what are the values (in a supplementary table perhaps) of the variables involved (mainly Species richness, Phylogenetic diversity and Phylogenetic species richness) that lead to these correlations.

Since my previous comments require a thorough revision of the text, or at least an extensive reply, I am not going to enumerate other minor details (errata or lack of references in the cited literature). Lastly, I provide an appreciation on Figure 3 and on Figure S1. In the first case the right ordinate axis can be entitled with number of species and in the second one it could be necessary a Figure caption (that I have not found in the text) where the complexity of it could be properly explained.

Reviewer 2 Report

This study aims to improve measurement of richness-focused phylogenetic diversity at the community (or assemblage) level. The novelty is the new metric, TILD, that is presented in the manuscript, and the empirical exploration of the patterns produced by this new measure in comparison with existing measures. I think this can make a publishable addition to the literature because the new measure is useful, even though it is so similar to Faith’s PD (hereafter just PD). However, unfortunately the current manuscript suffers horribly from circularity and artefact. I suggest that a fundamental rewrite is needed, taking a different approach to presenting the bits of the study that are useful.  I now explain why I say these things.

The fundamental problem is the circularity. This is perhaps best illustrated by close examination of the first paragraph of the Conclusions section, L369-77:

The first sentence is: ‘Our study has shown that phylogenetically-derived metrics of community lineage diversity outperform taxonomically-derived metrics.’  See also the abstract: ‘We find that phylogenetic metrics outperform taxonomic metrics’ (L6), and ‘The best metric is newly derived here’ in the next sentence.  And the start of the Discussion.

But on what basis is this judgment made? How do we know which measure is better? These questions leapt out at me as soon as I read the abstract - which does not answer it.  The answer eventually comes in the last paragraph of the Methods (from L180), and is that the authors produced another measure of CLD, which they used as the answer, against which to compare all 9 metrics ‘tested’. That is, the judgment was done according to the correlations of the metrics with this response variable. This response variable was ‘the number of lineages at different phylogenetic depths (in intervals of 5 Myrs between the present and the root of the seed plant phylogeny at 350 Ma).’ The conclusions about the performances of the metrics are mainly based on the mean Spearman’s rank correlation between each metric and the number of lineages at each 5Myr time-slice - averaged across all the time-slices, and shown as the rho values above the graphs in Fig. 4. In addition, the trends through phylogenetic time in the correlations are examined to draw conclusions about which metrics correlate best with the phylogeny in different periods of the evolutionary history of the taxa currently in a given assemblage.

This immediately raises the fundamental issue that whichever metrics most closely match the response variable (mathematically) will be the ones judged to be the best.

PD. According to L270-1, PD ‘is identical to the area beneath a raw (i.e., non-log-transformed) lineage through time plot.’  Also see L149-52: ‘If one constructs a lineage through time (LTT) plot for each community (sensu [18]), one can simply integrate the area under this curve as a measure of the total lineage diversity of the community over time. In fact, this integral is mathematically identical to the phylogenetic diversity of the community, when including the root branch.’  Note: I think it is important that the root branch is included in a sensible measure of phylogenetic diversity (see below).

The ‘answer’. The response variable is effectively just sampling the lineage through time (LTT) plot at 5Myr intervals.  So the response variable is an empirical estimate of PD! The claim that TILD and PD outperform others is an artefact of how the test was done.

TILD. The mathematical details and proofs are not given or cited in the manuscript – I think they should be. From the statements made in the manuscript, the new metric, TILD, seems to be very closely related to PD, mathematically, and therefore also to the response variable:

Both PD and TILD are very similar mathematically (see e.g. L270-1). The only difference between PD and TILD seems to be the log transformation, which is explained in L153-6; it is that PD is biased towards recent lineages when only extant species are included.  To address this bias, the Y-axis of the LTT plot is log-transformed before the integral (TILD) is calculated (L157-8). It may be that there is also another difference between PD and TILD: it is unclear from the manuscript whether the inclusion of the root is also a difference; I suspect not.

So apparently the ‘answer’ (response variable), PD and TILD are all very similar mathematically, differing only in a log transformation (PD vs TILD) and an estimate from time slices vs an integral. It is therefore little more than a mathematical tautology that PD and TILD are very strongly correlated (r = 0.95 according to Fig. S1).

Most importantly, it is circular that both PD and TILD are strongly correlated with the ‘answer’, which is just a way of estimating PD.  So I do not find it appropriate to write this work up as an empirical test of the metrics that finds that PD and TILD (and also PSR, which is also almost identical) ‘out-perform’ other metrics.  It is circular reasoning.

It is also effectively a tautology (i.e. an artefact of the methods) that TILD correlates less well with the response in more recent parts of the phylogeny.  The log-transformation within TILD was deliberately designed to diverge from PD nearer to the present. The same applies to the better correlation between the response and PD in more recent parts.  These results were all pre-determined by the methods used.

Coming back to the first paragraph of the Conclusions:

‘Therefore, molecular phylogenies should play a key role in informing conservation prioritisation schemes. Thus, the continued effort to sequence all species in the tree of life receives clear support from our study.’ (L370-2.)  This is again circular. The ‘answer’ was defined by molecular phylogenies, so there is no new evidence in this study with which to support such a statement.  It is a bit like saying that because bird, herp and mammal diversity measures correlate better with vertebrate diversity than plant diversity does, vertebrates should be used to inform conservation efforts.

The next sentence is: ‘As for recommended metrics, the phylogenetic diversity of communities is a valid measure of CLD, and we do not suggest that the many previous studies to use the metric are invalid.’  Again, this conclusion is not based on any new evidence.  The ‘answer’ was an estimate of PD, so PD correlated strongly with it.  One cannot conclude anything more than that.

The final sentence of the paragraph: ‘Rather, we stress that it is biased towards the diversity of recently derived lineages, and we urge consideration of the newly developed metric TILD (time integrated lineage diversity), which may be more representative of CLD over the full evolutionary history of the clade of study.’  I do not disagree with this statement, but it is about the reasoning used for TILD in the first place.  The authors should not imply that the empirical investigation has provided independent evidence of the value of the new metric.

The other paragraph of the Conclusions is also apposite. The results of the empirical work with the FIA tree data are already well known: there is more phylogenetic diversity when you have both angiosperms and conifers than when you only have one of these clades – when you use this sort of measure of phylogenetic diversity. This is admitted in e.g. L261. The lack of anything novel in the empirical results suggests to me that this should be a methods paper, not a research paper.

So where does this leave the manuscript? The authors have produced a new measure of phylogenetic diversity, albeit one that is very similar to an existing one.  PD is widely used, and I think the adjustment that TILD represents does have some use.  PD emphasises the more recently evolved taxa more than TILD does.  The two measures can be used in tandem to investigate nuances in data.

I advise that the authors reorganise the manuscript to:

1. Present TILD, including more details of its mathematics and clearer comparison to PD.

2. Use the FIA data to show how the patterns obtained from TILD differ from those of PD and the other existing measures.

This is not so very different in terms of content from what is already in the manuscript, but crucially it would remove the pretence of empirical validation - and therefore it would remove the circularity problem. The empirical ‘validation’ would instead be presented as empirical EXPLORATION, which I consider to be much more appropriate.

Perhaps the most surprising thing in the empirical results is that ‘CLD’ correlates slightly more strongly with TILD than with PD (mean rho of 0.77 vs 0.72). If I have understood the measures correctly, CLD estimates PD more than TILD because CLD and PD are both based on the untransformed y-axis of the LTT plot, while TILD is based on the log-transformed y-axis.  When only comparing ranks, the effect of the transformation may be reduced, but even so, I would expect the correlation between CLD and PD to be no lower than that between CLD and TILD.  As such, I would like to see a significance test done: are these two mean correlations significantly different? The simplest approach would be a paired-samples t-test (the pairs being the rho-values for each depth).

I also found it quite interesting in L209 and Fig. S1 that the TILD measure, while is very similar to PD and phylogenetic SR (r >= 0.94, though the units are different), the correlation with species richness is rather weaker (r = 0.59 vs 0.75 or 0.76). This suggests that the small amount of variation in TILD that is not correlated with PD is quite independent of species richness, which may be a useful feature, if the same is true in other datasets.

FURTHER COMMENTS

The comments below are largely copied from notes I took while reading the manuscript; I’m afraid that I don’t have time to turn them into flowing prose! Even so, they should be helpful for improving the manuscript.

Abstract omits key information.  First, it states that the term lineage diversity can refer to genetic lineages [within species] as well as deeper lineages than species, but fails to state that in this paper you are only looking at the latter and not the former.

It also fails to state that the manuscript is focused only on the richness dimension of phylogenetic diversity – a narrower focus than implied by the title and abstract.

Second, the abstract does not state how you judged which metric performs best. You correctly say that it is not clear how to best quantify CLD, but then contradict yourselves by then saying that you tested how well several metrics measure CLD.  The contradiction, of course, is that if we don’t know how to derive the answer (‘true’ CLD) then we don’t know how to judge which metric measures it better.  See also L87-8.

(Some reflections on whether this sort of approach of using accumulated lineage age is appropriate for concluding conservation value; note that this sort of thinking is why I think it is important to include the root branch in a sensible measure of phylogenetic diversity. This is food for thought rather than suggestions for specific changes to the manuscript:

In L38-9 you make a statement that is contentious, but which is a key assumption underlying this manuscript: ‘However, Linnean taxonomic ranks are not ‘natural’ in the sense that they do not directly correlate to any precise evolutionary age. age.’  While I (and probably most scientists) agree that Linnaean many taxonomic ranks are not necessarily ‘natural’, the notion that a natural rank must correlate to a precise evolutionary age is much less clear-cut.

The next sentence is ‘Some clades of a given taxonomic rank may actually be younger than clades of a putatively lower taxonomic rank.’  My response: so what? Are we not trying to measure diversification with lineage diversity, rather than simply accumulated age?

Similarly ‘could mean that the community of Pinus has greater conservation value in terms of encompassing greater total evolutionary history’ (L47-8) is suggesting that the only thing that matters about evolutionary history is the ages of lineages. Yet we know that stasis can happen for very long periods in many places, while diversification can happen very fast in some circumstances.

àWhat is the common currency of lineages?  Number of genetic substitutions?

The key advance here is in time-integrating dated phylogenies to produce a single measure of phylogenetic diversity.  But this is built on a number of key assumptions. Basically, it seems like a slightly better measure than existing ones, within the same set of restrictive assumptions.

So the research is within a very specific self-imposed remit.  Within that, it is OK.)

L100: the phylogenetic data used were ultrametric, and so lose information about the rate of accumulation of genetic diversity within lineages through time.

I find it a shame that the study is restricted to ultrametric trees, and thus the emphasis only on age.  At the very least this needs discussion, but better would be to expand to also using non-ultrametric trees, which would measure genetic difference as well as age.

L70 ‘it seems plausible that Community A has more lineage diversity than Community C, even though Community C has more species.’  Only under the restrictive set of assumptions involved!  Opportunity missed, perhaps.  Note that most speciation events do not actually reset the clock: the genetic lineage diversity within one branch (accumulated over long periods) is not automatically lost at the moment of speciation…

L71-80: much better.

In the FIA dataset, were plantations removed?  According to L197, there are something like 20,000 plots with only one or two tree species, suggesting plantations may be in the dataset.  If they were not removed, then the maps (Fig. 5) are affected by this, with important implications with respect to  objective 2, L88-90.

In deriving TILD, why was the transformation log and not, for example, square root?  (See L156-8.) I guess that this is to counteract the exponential increase mentioned in L155, in which case there is a theoretical reason for this particular transformation. If so, this should be explained here.  If not, then details should be given of other transformations tried, and why log was better.

SPECIFIC COMMENTS

Be careful with blanket statements about conceptual equivalence. For example ‘many lineages, and therefore more evolutionary history’ (L24) equates number of lineages and amount of evolutionary history.  I would say that evolutionary history is a broader concept than number of lineages, and that although the two phenomena are positively correlated, the correlation is far from perfect.

In the following statement surely the main point is that the term lineage diversity is used in both micro- and macro-evolution. ‘The majority of times the term lineage diversity has been used in the literature, it refers to the number of diverged genetic lineages within species (e.g., [5–9]), but it has also been used for deeper evolutionary levels (e.g., [10–14])’ (L31-3). I do not find the pseudo-quantification in the sentence very helpful, and it distracts from the main point. 

Fig 1 says ‘Time units are arbitrary’, yet they are stated in million years before present. How is that any more arbitrary than the mock phylogenies?

Beware of hyperbole.  For example, ‘unparalleled community phylogenetic dataset’ (L87).  Very similar datasets exist for Canada and Mexico (similar size of dataset and almost identical data collection protocols, but with some advantages compared with the freely available FIA dataset, which is location-fudged because of land ownership issues specific to the USA). There may be other similar datasets, too.  The US dataset is more accessible than others, but it is not unparalleled.

There is a problem with the reference given in L101: ‘we combined the temporally calibrated ultrametric phylogenies … from Ma textitet al. 2016.’  The word ‘textit’ appears several times in the manuscript – always problematic.  I guess the reference given is Ma et al 2016, which is reference 48 in the list.  [Fig.2 legend confirms this.]

Figure 2 is not referred to in the text until 2 pages after the figure, and after Fig 3 is referred to.

Is Fig.2 complete?  There should be almost exactly 200 angiosperm tree species in the phylogeny, plus the conifers.  It is very hard to tell, but there seem to be fewer than 200.

Isn’t it ‘rarefied’, not ‘rarefacted’?

‘The most species-rich communities have anywhere from 9 to 17 species per 25 stems’ (L198). Bizarre statement! The most species rich have 17!  I only figured out that the authors mean the top 10% of values when I read Fig 3 caption. Needs addressing in the text.

‘Considering these species-rich communities, we see a wide range of variation in when these communities accumulated lineage diversity’ (L200-1). This makes it sound like the diversification was all in situ, with no dispersal!

‘The second-largest step-change in the number of lineages present in communities occurs between 5 Ma and the present’ (L206-7). This is presumably the artefact of using only extant species.

Fig. S1: what are the graphs on the diagonal?